SHAPS-C: the Snaith-Hamilton pleasure scale modified for clinician administration

Ameli Rezvan 1 Rezvan.Ameli@NIH.GOV
Luckenbaugh David A. 1
Gould Neda F. 2
Holmes M. Kathleen 3
Lally Níall 1 4
Ballard Elizabeth D. 1
Zarate Carlos A. Jr 1
1 Experimental Therapeutics and Pathophysiology Branch (ETPB), National Institutes of Mental Health , NIH, DHHS, Bethesda, MD , USA
2 Department of Psychiatry and Behavioral Sciences, Johns Hopkins University School of Medicine , Baltimore, MD , USA
3 Department of Clinical Psychology, St. John’s University , Jamaica, NY , USA
4 Institute of Cognitive Neuroscience, University College London , London , UK
Black Kevin
Electronic publication date: 2014 Jun 17
Publication date: 2014
Volume: 2
Electronic Location ID: e429
Received 2014 Mar 17; Accepted 2014 May 26
Copyright: © 2014 Ameli et al.
Copyright year: 2014
Copyright holder: Ameli et al.
License: This is an open access article distributed under the terms of the Creative Commons Attribution License, which permits unrestricted use, distribution, reproduction and adaptation in any medium and for any purpose provided that it is properly attributed. For attribution, the original author(s), title, publication source (PeerJ) and either DOI or URL of the article must be cited.
License URL: https://creativecommons.org/licenses/by/4.0/

Keywords: Self-assessment, Anhedonia, Depression, Clinician administered

Funding: Intramural Research Program of the National Institute of Mental Health National Institutes of Health (NIMH-NIH) The authors gratefully acknowledge the support of the Intramural Research Program of the National Institute of Mental Health, National Institutes of Health (NIMH-NIH) and thank the 7SE research unit of the NIMH-NIH for their support. The funders had no role in study design, data collection and analysis, decision to publish, or preparation of the manuscript.

==============================
Anhedonia, a diminished or lack of ability to experience and anticipate pleasure represents a core psychiatric symptom in depression. Current clinician assessment of anhedonia is generally limited to one or two all-purpose questions and most well-known psychometric scales of anhedonia are relatively long, self-administered, typically not state sensitive, and are unsuitable for use in clinical settings. A user-friendly tool for a more in-depth clinician assessment of hedonic capacity is needed. The present study assessed the validity and reliability of a clinician administered version of the Snaith-Hamilton Pleasure Scale, the SHAPS-C, in 34 depressed subjects. We compared total and specific item scores on the SHAPS-C, SHAPS (self-report version), Montgomery-Åsberg Depression Rating Scale (MADRS), and the Inventory of Depressive Symptomatology-Self Rating version (IDS-SR). We also examined construct, content, concurrent, convergent, and discriminant validity, internal consistency, and split-half reliability of the SHAPS-C. The SHAPS-C was found to be valid and reliable. The SHAPS and the SHAPS-C were positively correlated with one another, with levels of depression severity, as measured by the MADRS, and the IDS-SR total scores, and with specific items of the MADRS and IDS-SR sensitive to measuring hedonic capacity. Our investigation indicates that the SHAPS-C is a user friendly, reliable, and valid tool for clinician assessment of hedonic capacity in depressed bipolar and unipolar patients.

Introduction

A diminished or lack of ability to experience or anticipate pleasure or anhedonia, and its assessment is central to understanding and treating depressive states (Hasler et al., 2004; Klein, 1974; Robinson et al., 2012; Spijker et al., 2001a; Spijker et al., 2001b; Treadway & Zald, 2011). Research has indicated a distinct neurobiological difference between consummatory and anticipatory pleasure; evidence suggests that the latter is strongly aberrant in depression (Treadway & Zald, 2011). The National Institutes of Mental Health Research Domain Criteria (NIMH, RDoC; Insel et al., 2010) considers anhedonia a central construct for both better understanding of depression and discovery of more effective treatments (Cuthbert, 2014). Several self-rated scales for the assessment of hedonic capacity have been published, including the Chapman Revised Physical and Social Anhedonia Scales (CRPAS/CRSAS; Chapman, Chapman & Raulin, 1976), the Fawcett Clark Pleasure Scale (FCPS; Fawcett et al., 1983), and Snaith-Hamilton Pleasure Scale (SHAPS; Snaith et al., 1995). The latter is a 14-item, self-rated user-friendly measure that addresses shortcomings of previous measures, such as length, state versus trait sensitivity, and the relatively culture free nature of questions (Snaith et al., 1995), and has been further validated in independent samples since the original study (Franken, Rassin & Muris, 2007; Leventhal et al., 2006; Nakonezny et al., 2010). Furthermore, Research Domain Criteria (RDoC; Insel et al., 2010) included the SHAPS as a potential measure of ‘sustained responsiveness to reward’, which is related to anhedonia. Reliable and valid measurement of hedonic capacity will only increase in importance as RDoC is incorporated into future research.

The SHAPS is a self-rated tool. The value of self-assessments in depressive states has been called into question (Corruble et al., 1999; Prusoff, Klerman & Paykel, 1972a; Prusoff, Klerman & Paykel, 1972b). While the effective use of self-assessment has been reported (Rush et al., 1986), severity of illness, presence of personality disorders, instructions, motivation, and mood-dependent memory are among the factors that can compromise the objectivity of self-assessments (Blaney, 1986; Corruble et al., 1999; Prusoff, Klerman & Paykel, 1972b). Emerging evidence from both neuroimaging work and behavioral studies suggest that neural/behavioral responses for people with severe mental disorders (e.g., schizophrenia) are different from their self-reported responses. For example, there are neuroimaging data that suggest patients with schizophrenia show intact patterns of increased ventral striatum responses to reward receipt itself (Dowd & Barch, 2012), although they tend to report reduced ability of experiencing pleasure according to a self-report measure of anhedonia (Kring & Moran, 2008). Furthermore, some have suggested that a complete assessment of depression should include both clinician-rated and self-report measures since each uniquely contribute to the prediction of treatment outcome (Uher et al., 2012).

Based on the initial promise of the SHAPS, we modified this scale for use as a clinician-administered tool, SHAPS-C, by adding specific item wording, instructions, and probe questions, as well as modification of the scoring. Care was taken to phrase the questions such that both the consummation and the anticipatory aspects of anhedonia could be assessed. The SHAPS-C includes the same 14 areas of hedonic experience as the SHAPS. SHAPS items are scored 0 or 1. Items on the SHAPS-C are scored from 1 to 4 (1 = Lots of pleasure, 4 = No pleasure) to allow for greater score variability (Franken, Rassin & Muris, 2007; Liu et al., 2012). The inclusion of “lots of pleasure” which is scored 1, can also allow the investigation of high moods should that be of assessment interest particularly in bipolar conditions (SHAPS-C, see Supplemental Information).

The construct, content, and face validity of the SHAPS-C stem from the fact that it is closely modeled after the SHAPS and explores identical areas of hedonic capacity. We further assessed the concurrent validity of the SHAPS-C by examining its relationship to the SHAPS in a group of unipolar and bipolar depressed patients. The convergent validity was assessed by examining the relationship between the SHAPS-C and specific items of the MADRS and IDS-SR assessing hedonic capacity. Similarly, we assessed its discriminant validity by looking at items from the Montgomery-Asberg Depression Rating Scale (MADRS; Montgomery & Asberg, 1979) and Inventory of Depressive Symptomatology-Self Rating (IDS-SR; Trivedi et al., 2004) that are not presumed to be directly related to hedonic capacity. In addition, we examined the reliability of the SHAPS-C by assessing its internal consistency and split-half reliability.

Materials and Methods

We studied 34 depressed subjects (18 males) with a mean age of 46.7 years (SD = 10.4, range 24–63) who participated in depression studies at the National Institute of Mental Health (NIMH), Bethesda, MD, under Institutional Review Board approved protocols (01-M-0254) including written informed consent. Subjects were diagnosed based on a best estimate diagnostic procedure that included psychiatric interview, assessment by the Structured Clinical Interview for the Diagnostic and Statistical Manual of Mental Disorders (DSM-IV-TR) patient edition (SCID I/P; First et al., 2002), and interview of family members as well as review of past history and records as indicated. Subjects who met criteria for current Major Depressive Disorder (MDD; n = 21) or Bipolar Disorder (BD, n = 13) in the depressive phase participated. Subjects with current psychosis, cognitive impairment, unstable medical conditions, or acute suicide risk were excluded. We also excluded manic or hypomanic subjects (n = 2). Although the SHAPS-C can measure increased pleasure, such a low number of subjects did not justify inclusion into the study. All subjects completed the SHAPS-C, SHAPS, MADRS, and the IDS-SR (Table 1). The same clinician administered the MADRS and the SHAPS-C.

Table 1 Demographic and clinical characteristics of study sample.

	N (%)	
Diagnosis		
Major depressive disorder	21 (62)	
Bipolar disorder	13 (38)	
Gender (Male)	18 (53)	
Race (Caucasian)	21 (62)	
	Mean (SD)	
Age	46.7 (10.4)	
SHAPS-C total	41.9 (7.2)	
SHAPS total	6.5 (4.3)	
IDS-SR		
Total	43.5 (12.0)	
General interest	2.1 (0.9)	
Capacity for pleasure/enjoyment	1.8 (0.7)	
MADRS		
Total	32.7 (6.3)	
Inability to feel	3.8 (1.0)	
Notes.

SHAPS-C Snaith-Hamilton Pleasure Scale-Clinician Administered

IDS-SR Inventory of Depressive Symptomatology-Self Rating

MADRS Montgomery-Åsberg Depression Rating Scale

SHAPS Snaith-Hamilton Pleasure Scale

Pearson correlations were calculated to better understand the concurrent validity of the SHAPS-C. MADRS Inability to Feel (item 8) and IDS-SR General Interest (item 19) and Capacity for Pleasure and Enjoyment (item 21) were examined in relationship to the SHAPS-C total scores. Similarly, the discriminant validity of the SHAPS-C was assessed by the level of correlation between MADRS Concentration (item 6), Energy (item 7), and Pessimism/Guilt (item 9) and IDS-SR Concentration (item 15), Outlook Towards Self (item 16), Energy (item 20), and Somatic Concerns (item 25), items that are not presumed to be directly related to hedonic capacity. Significance was evaluated at p < .05, two-tailed. To have 80% power to demonstrate a correlation of at least r = .50, a minimum of 26 cases were required; 34 cases yielded 90% power. Finally, Cronbach’s alpha and the Spearman–Brown coefficient were used to examine the internal consistency, and split-half reliability of the SHAPS-C, respectively.

Results

The mean scores for the SHAPS-C, SHAPS, IDS-SR (total), and MADRS (total) for the study sample were 41.9 (SD = 7.2), 6.5 (SD = 4.3), 43.5 (SD = 12.0), and 32.7 (SD = 6.3), respectively (Table 1). This suggests a moderate to severely depressed sample.

The SHAPS-C was internally consistent (Cronbach’s α = .90). Removing individual items did not change the internal consistency substantially in either direction. The Spearman–Brown split-half reliability was .90. In addition to evidence for the reliability of the SHAPS-C, we also found support for the SHAPS as an internally consistent measure (Cronbach’s α = .88, Spearman Brown = .93).

The SHAPS-C was positively correlated with the SHAPS (r = .85, p < .001). Figure 1 illustrates this relationship and shows that patients had the full range of scores on the SHAPS, but they did not reach the lower levels of the SHAPS-C. This was expected since the lowest scores on the SHAPS-C would indicate higher than normal levels of pleasure which is not expected in a group of moderate to severely depressed patients. Given the overlap in the content of the questions for these scales, we examined the relationships between corresponding items. Spearman correlations were used due to the short range of values for the items. The correlations ranged from .37 to .73 with 12 of 14 items having correlations over .50.

Figure 1 Association between SHAPS-C and SHAPS.

A strong positive linear relationship between the clinician administered SHAPS-C and the self-administered SHAPS is apparent.

Figure 2 Association between the SHAPS-C and the MADRS.

A strong positive relationship is visible between the clinician administered SHAPS-C scale, which assess anhedonia, and the clinician administered MADRS, which assesses general depressive symptomatology.

Table 2 shows the relationships between the anhedonia (SHAPS-C, SHAPS) and depression scales (MADRS, IDS-SR) (see Fig. 2). As predicted, the SHAPS-C and SHAPS totals were significantly correlated with MADRS Inability to Feel (item 8) and IDS-SR General Interest (item 19) and Capacity for Pleasure or Enjoyment (item 21). These relationships suggest the convergent validity of the SHAPS-C. Interestingly, the correlation between hedonic capacity and mood ranged from low to moderate indicating that mood and hedonic capacity could be considered as relatively independent constructs. Specifically, the correlation between SHAPS and SHAPS-C totals were less than .5 for MADRS Apparent Sadness (item 1) and Reported Sadness (item 2), as well as with IDS-SR Sad Mood (item 5) and Mood Variation (item 9). Only the relationship with IDS-SR Mood reactivity (item 8) and SHAPS-C was just over .5. The SHAPS-C and SHAPS totals were not significantly correlated with MADRS Concentration (item 6), Energy (item 7), or Pessimism/Guilt (item 9), nor with the corresponding items of the IDS-SR Concentration (item 15), Energy (item 20), or Outlook Towards Self (item 16). These non-significant associations support the discriminant validity of the SHAPS-C.

Table 2 Correlations between the SHAPS, SHAPS-C, IDS-SR, and MADRS, and specific scale items.

	SHAPS	SHAPS-C	
	r	p	95% CI	r	p	95% CI	
SHAPS-C	0.85	<.001	0.71	0.92					
IDS-SR									
Total	0.52	0.003	0.20	0.74	0.55	0.001	0.25	0.76	
Item 5 (Sad mood)	0.34	0.07	−0.02	0.62	0.47	0.007	0.15	0.70	
Item 8 (Mood reactivity)	0.38	0.04	0.03	0.65	0.55	0.001	0.25	0.75	
Item 9 (Mood variation)	−0.21	0.27	−0.52	0.16	−0.25	0.17	−0.55	0.11	
Item 15 (Concentration)	0.16	0.38	−0.20	0.49	0.15	0.40	−0.21	0.48	
Item 16 (Outlook towards self)	−0.19	0.32	−0.51	0.18	−0.13	0.48	−0.46	0.23	
Item 19 (General interest)	0.39	0.03	0.04	0.66	0.48	0.006	0.15	0.71	
Item 20 (Energy)	0.29	0.12	−0.08	0.58	0.31	0.08	−0.04	0.60	
Item 21 (Capacity for pleasure
or enjoyment)	0.54	0.002	0.23	0.75	0.69	<.001	0.45	0.84	
Item 25 (Somatic concerns)	0.19	0.31	−0.18	0.51	0.22	0.23	−0.14	0.53	
MADRS									
Total	0.52	0.003	0.20	0.73	0.56	0.001	0.27	0.76	
Item 1 (Apparent sadness)	0.23	0.21	−0.13	0.53	0.48	0.005	0.16	0.70	
Item 2 (Reported sadness)	0.44	0.01	0.11	0.69	0.49	0.004	0.18	0.72	
Item 6 (Concentration)	0.16	0.40	−0.20	0.48	0.19	0.29	−0.16	0.50	
Item 7 (Energy)	0.12	0.51	−0.24	0.45	0.21	0.23	−0.14	0.52	
Item 8 (Inability to feel)	0.48	0.006	0.15	0.71	0.53	0.002	0.22	0.74	
Item 9 (Pessimism/guilt)	0.06	0.75	−0.30	0.40	0.01	0.95	−0.33	0.35	
Notes.

CPES Snaith-Hamilton Pleasure Scale-Clinician Administered

SHAPS-C Snaith-Hamilton Pleasure Scale-Clinician Administered

IDS-SR Inventory of Depressive Symptomatology Self Rating

MADRS Montgomery Asberg Depression Rating Scale

SHAPS Snaith-Hamilton Pleasure Scale

Discussion

In-depth measurement of hedonic capacity along with the measurement of mood and behavior is important in depression treatment studies (Boyer et al., 2000). Self-administered assessments may not be sufficient, particularly in severe psychiatric conditions. In addition, for a complete assessment of depression both clinician-rated and self-report measures may be included since each can uniquely contribute to the prediction of treatment outcome (Uher et al., 2012). This study introduces the SHAPS-C, a clinician administered version of the SHAPS, and demonstrates its internal consistency, split-half reliability, and convergent and discriminant validity with the SHAPS, MADRS, and IDS-SR in a group of depressed patients for assessment of anhedonia. The SHAPS-C was strongly positively correlated with the original SHAPS and with specific hedonic items from the MADRS and IDS-SR, but not non-hedonic questions. The high correlation between the SHAPS and SHAPS-C suggests they tap into the same construct. However, the size of the correlation indicates that about a third of the variance (r2 = .67) from one is not explained by the other which may point to the uniqueness of a clinician measure. However, it should be noted that we utilized internal consistency and split-half to assess the reliability of SHAP-C, future studies may also want to consider test-retest reliability for a more complete assessment of reliability.

Similar correlations between the MADRS and SHAPS were reported in the original study by Snaith et al. (1995) although their original sample of 46 patients was not limited to depressed patients and included mixed psychiatric disorders that displayed anhedonia. The similarity between results from the initial SHAPS study and the current study suggest that mood and hedonic capacity could be considered separate constructs and closer attention should be paid to the assessment of hedonic capacity. In addition, laboratory findings suggest that underlying neurobiological and neuropsychological substrates for anhedonia may be useful in clarifying relevant endophenotypes of depression related to anhedonia (Gottesman & Gould, 2003; Hasler et al., 2004; Pizzagalli et al., 2009; Pizzagalli, Jahn & O’Shea, 2005). Hedonic tone may assist in elucidating links and differentiations among various psychiatric disorders (Snaith et al., 1995) including bipolar conditions and depression subtypes.

The present study has several limitations that future studies should address. While the sample size of 34 yielded 90% power for the study, a larger sample size will be needed to confirm the current findings. Non-significant correlations intended to demonstrate divergent validity could be significant in a larger study which would cloud the arguments for the uniqueness of the SHAPS-C. Future studies reporting on validity and reliability of the SHAPS-C in larger clinical samples will help to clarify the usefulness of this measure and provide more accurate estimates of the relationships between the SHAPS-C and other measures. The subject population of the current study was limited to moderate to severely depressed patients, the performance of the SHAPS-C in euthymic or mildly depressed patients remains to be studied. Including euthymic or mildly depressed patients would help us to better understand the utility of SHAPS-C across the full spectrum of depression and hedonic capacity. There were no control groups in the current study. The addition of various control groups in future studies will further enrich the interpretation of current findings. In particular, assessment of hedonic capacity not only in depressive states but also in manic or hypomanic states will require a tool for assessment in both directions. SHAPS-C’s scoring and questions are designed such that a bidirectional assessment can take place. In addition, the use of the measure with control subjects as well as a wider range of patient groups could provide normative information to establish normal and pathological levels of hedonic capacity. Finally, since the same clinician administered the clinician measures, the potential for clinician bias could have increased the correlation between these measures which means the estimates of the relationships could be overstated. However, this could be true for evaluations of convergent and divergent validity.

In sum, exploration of hedonic capacity in diagnostic, clinical, and neurobiological investigations requires valid and reliable tools. Given the controversy of self-assessments in severe psychiatric disorders, including depression, and possible unique contribution of self-assessments and clinician-assessments for prediction of outcome, the availability of a user-friendly clinician-administered tool for assessment of anhedonia is of potential value. We propose that SHAPS-C could be considered such a tool.

Supplemental Information

Supplemental Information SHAPS-C tool: the Snaith-Hamilton Pleasure Scale modified for clinician administration

A 14 item scale modified based on the original self administered SHAPS tool. This measure permits the assessment of anhedonia via clinician administered questions.

Click here for additional data file.

Additional Information and Declarations

Competing Interests

Author Contributions

Human Ethics

None of the investigators in this study have a possible conflict of interest, financial or otherwise.

Rezvan Ameli conceived and designed the experiments, performed the experiments, analyzed the data, wrote the paper, prepared figures and/or tables, reviewed drafts of the paper.

David A. Luckenbaugh analyzed the data, wrote the paper, prepared figures and/or tables, reviewed drafts of the paper.

Neda F. Gould and M. Kathleen Holmes performed the experiments, wrote the paper, reviewed drafts of the paper.

Níall Lally wrote the paper, prepared figures and/or tables, reviewed drafts of the paper.

Elizabeth D. Ballard and Carlos A. Zarate Jr wrote the paper, reviewed drafts of the paper.

The following information was supplied relating to ethical approvals (i.e., approving body and any reference numbers):

National Institute of Mental Health (NIMH), Bethesda, MD Institutional Review Board approved the protocol (01-M-0254) and all participants provided written informed consent.

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
