# Peer review of "SHAPS-C: the Snaith-Hamilton pleasure scale modified for clinician administration"

_PeerJ, doi:10.7717/peerj.429_

## Round 0.1 · original submission · Minor Revisions

This is an interesting report on a potentially useful rating scale for anhedonia from a capable group. The authors attach the full text of the new scale to the article, which will help others (as well as probably boost your citations). I believe you can address the most important concerns of the reviewers without studying additional patients.

1. The most important concern raised by the reviewers relates to your sample size. As I believe you intended to communicate in lines 101-106, the primary concern with your sample size is actually whether you missed small-to-moderate true correlations that would reduce the evidence for discriminant validity. For instance, the correlation with the IDS-SR Energy item may well be real (p=.08 in this sample). Rather than the vague comment currently found in lines 170-173, please address this specific limitation. In fact, the optimal way to address it would be to report confidence intervals for the r values currently given in lines 136-143. While you're at it, lines 132-142 would be improved by moving the numbers to a table.

2. A separate but related limitation of the sample studied is: what do you lose by only including moderately to severely depressed patients, and no mildly depressed patients or euthymic controls? (cf. Fig. 2) Please address this as appropriate.

3. Were SHAPS-C and MADRS done by the same clinician at the same interview? If so, note the consequences of this as a limitation of the study in Discussion.

Finally, please consider and address in your reply the additional points listed below, along with all other comments from the reviewers. (Except, don't stress about the formatting issues mentioned; the journal production team can help with that after you upload your final files.)
4. Line 74 -- Face validity of a purported measure of anhedonia should be judged by comparing it to a reference standard assessment of anhedonia in its best-known context (e.g. expert judgment of anhedonia in major depression). As one possible step in that direction, the SCID gives data on whether the current depressive episode meets research criteria for melancholic type. I would be very interested if you could include information on whether (or to what extent) the SHAPS-C corresponds to melancholia.
5. Line 83 -- Reliability is more relevant when measured by test-retest assessments than by split-item comparisons. Discuss this in limitations.
6. It strikes me as odd to think of validating a clinician rating against a self-report measure, rather than vice versa.

Reviewer 1 ·

Basic reporting

• Clear and unambiguous writing, short and concise
• Sufficient introduction and background
• Relevant figures, however they should be improved. Now it looks as Fig 1 and 2
are “copy pasted” from the output of a statistical software. Also, Figure captions
were missing
• The tables should also be improved, it looks like the columns have been
“jumping” (Table 1), and abbreviations could be used for Table 2, to make the
columns less wide.

Experimental design

• The research aim is meaningful and clearly defined
• The methodology is to a high standard, although too few subjects which raises
alarming questions for the validity of the study
• Methods are described with sufficient information

Validity of the findings

• The data on which the conclusions are based on are provided
• The conclusions are appropriately stated, although there is a power issue. To
validate or introduce a new measurement based on only 34 subjects is not
enough, even though the data so far looks very promising.
• The authors state (line 105-106): “To have 80% power to demonstrate a
correlation of at least r=.50, a minimum of 26 cases were required; 34 cases
yielded 90% power”. Why aim for a correlation of r=.50? They should aim higher
if the object of the study is to introduce a new scale. At least 100 subjects are
required for this, especially when calculating Cronbach’s alpha for specific items.

Additional comments

To introduce a new, easily administered, scale for evaluation of anhedonia is very meaningful and something that is needed and potentially can improve the treatment of depression and anhedonia symptoms in other psychiatric disorders. It is also very good that this scale is aimed to be administered by a clinician and that it is not only self-report.

Your data looks very promising. However, my concern lies in your small sample size. I encourage you to add the number of participants, around 100, since I really think this scale can make a difference and the introduction of it deserves a larger sample size in order for its validation and relaiability.

Reviewer 2 ·

Basic reporting

No comments

Experimental design

No comments

Validity of the findings

No comments

Additional comments

This topic is timely and authors followed appropriate methods to provide evidence supporting validity and reliability of the SHPS. It would be good to add more elaborated explanations why interview-based measure of the SHPS is needed or important in the Introduction and Discussion sections as well. For example, in the line of 55-60 in the Introduction, authors mentioned about general limitations of using self-report measure (e.g., severity of illness, presence of personality disorders). It should be noted that important point for the use of interview-based measure rather than self-report measure comes from emerging evidence from both neuroimaing work and behavioral studies suggesting that neural/behavioral responses for people with severe mental disorders (e.g., schizophrenia) are different from their self-reported responses. For example, there are empirical evidence from neuroimaging data suggesting that patients with schizophrenia show intact pattern of increased ventral striatum responses to reward receipt itself (e.g. Dowd and Barch, 2012), although they tend to report reduced ability of experiencing pleasure according to self-report measure of the Chapman anhedonia (look up Kring and Moran, 2008 for relevant review).

References:
Dowd, E. C., & Barch, D. M. (2012). Pavlovian reward prediction and receipt in schizophrenia: relationship to anhedonia. PLoS One, 7(5), e35622.
Kring, A. M., & Moran, E. K. (2008). Emotional response deficits in schizophrenia: insights from affective science. Schizophr Bull, 34(5), 819-834

---

## Round 0.2 · Minor Revisions

I still feel these data are important, but surprisingly, the revised manuscript does not follow a couple of important instructions in the first decision letter. These points can still be corrected easily. If the intent of my requests is not clear, please email me for clarification before re-submitting. Specifically:

#. Please address the specific power limitation described in Editor point #1.
#. Also from Editor point #1, please add confidence intervals to the table for the r values used to support discriminant validity (or you can just add C.I.'s for all the r values in the table, if you prefer).
#. The reply to Editor point #2 is true, but I was hoping for something more specific. Imagine for instance that you had studied an additional dozen or so patients with MADRS scores distributed evenly from 0 to 20, and imagine that the SHAPS-C score turned out to be 32 for every one of them. Or 14. Of course either of these outcomes is unlikely, but either would substantially change your interpretation of Figure 2. Anyway, please consider changing the added text to say something more like, “The subject population of the current study was limited to moderate to severely depressed patients, so the performance of the SHAPS-C in euthymic or mildly depressed patients remains to be studied.”
#. For Editor point #3, the intent was to ask you to acknowledge that significant correlations between the MADRS and the SHAPS-C must be taken with a grain of salt since they were rated by the same person from the same examination. Please add such an acknowledgment to Discussion in place of the comment about "clinician bias."
#. Editor point #4: Please address the following in your reply letter: did you not do the melancholia section on the SCID? did you use a SCID version that omits it? do you no longer have access to the original SCID data? or do you have the raw data, but you judged it too onerous to go back to the SCID melancholic specifier section to figure out which subjects had it?

---

## Round 0.3 · accepted · Accept

The authors have more accurately summarized the potential limitations of this initial study.

You will probably want to add the word "so" after the comma below.
"The subject population of the current study was limited to moderate to severely depressed patients, the performance of the SHAPS-C in euthymic or mildly depressed patients remains to be studied."